# Toolbox Accelerating Glycomics (TAG): Glycan Annotation from MALDI-TOF MS Spectra and Mapping Expression Variation to Biosynthetic Pathways

**DOI:** 10.3390/biom10101383

**Published:** 2020-09-28

**Authors:** Nobuaki Miura, Hisatoshi Hanamatsu, Ikuko Yokota, Kazue Okada, Jun-Ichi Furukawa, Yasuro Shinohara

**Affiliations:** 1Division of Bioinformatics, Niigata University Graduate School of Medical and Dental Sciences, 1-757 Asahimachi-dori, Chuo-ku, Niigata 951-8510, Japan; 2Department of Advanced Clinical Glycobiology, Faculty of Medicine and Graduate School of Medicine, Hokkaido University, Kita21, Nishi11, Kita-ku, Sapporo 001-0021, Japan; h_hanamatsu@med.hokudai.ac.jp (H.H.); iyokota@med.hokudai.ac.jp (I.Y.); okada@soyaku.co.jp (K.O.); j_furu@med.hokudai.ac.jp (J.-I.F.); 3Department of Pharmacy, Kinjo Gakuin University, Nagoya 463-8521, Japan

**Keywords:** glycomics, automated annotation, MALDI-TOF MS, biosynthetic pathways, glycan diversity, informatics, expression variation

## Abstract

Glycans present extraordinary structural diversity commensurate with their involvement in numerous fundamental cellular processes including growth, differentiation, and morphogenesis. Unlike linear DNA and protein sequences, glycans have heterogeneous structures that differ in composition, branching, linkage, and anomericity. These differences pose a challenge to developing useful software for glycomic analysis. To overcome this problem, we developed the novel Toolbox Accelerating Glycomics (TAG) program. TAG consists of three units: ‘TAG List’ creates a glycan list that is used for database searching in TAG Expression; ‘TAG Expression’ automatically annotates and quantifies glycan signals and draws graphs; and ‘TAG Pathway’ maps the obtained expression information to biosynthetic pathways. Herein, we discuss the concepts, outline the TAG process, and demonstrate its potential using glycomic expression profile data from Chinese hamster ovary (CHO) cells and mutants lacking a functional *Npc1* gene (*Npc1* knockout (KO) CHO cells). TAG not only drastically reduced the amount of time and labor needed for glycomic analysis but also detected and quantified more glycans than manual analysis. Although this study was limited to the analysis of *N*-glycans and free oligosaccharides, the glycomic platform will be expanded to facilitate the analysis of *O*-glycans and glycans of glycosphingolipids.

## 1. Introduction

Cell surfaces are coated with a variety of intricately arranged glycoconjugates such as glycoproteins and glycolipids, and glycosylation is thought to be essential for maintaining homeostasis in mammalian cells [1,2]. As omics approaches such as transcriptomics and proteomics have contributed enormously to our understanding of various biological processes in the postgenomic era, rapid and precise analysis of cellular glycomics is attracting a lot of attention. Many useful software packages have been developed for transcriptomics and proteomics that mine useful information from next-generation sequencer and mass spectrometry (MS) data [3,4,5,6]. However, similar programs for glycomic analysis are scarce, which prevents the expansion of the field. Although glycan sequences and structures can be determined from tandem MS (MS/MS) data using dedicated software [7], tools for annotating glycan signals on mass spectra via large-scale quantitative analysis are not currently available. Glycome profiles differ between species, tissues, and cells, which makes it difficult to determine the number of glycans and their structures [8]. Distinguishing genuine glycan signals from contaminating signals can be difficult. There are various methods for modifying carboxylic acid of sialic acid and for derivatizing the reducing termini of glycans for glycomic analysis. Thus, glycans are empirically annotated from raw mass data, and quantitative analysis calculations are often carried out manually using spreadsheet software such as Microsoft Excel. Such manual approaches are too time-consuming and labor-intensive for large-scale glycomic analysis. Furthermore, manual approaches carry an increased risk of errors such as improper annotation, missing data, and improper sorting of results between analyses. Lack of a comprehensive glycan list and obtaining a non-standard resultant format owing to manual analyses can cause glycomics to be difficult.

Due to these difficulties, glycomic research is somewhat behind DNA- and protein-based approaches. Therefore, the development of appropriate software for automated data analysis, even if only partially automated, could accelerate glycomic research. Indeed, the automation of glycan annotation is a major goal in this field. In the present study, we developed novel software named Toolbox Accelerating Glycomics (TAG). TAG is composed of three functions that assist glycomic analysis: TAG List automatically creates a semi-comprehensive glycan list for database searching during glycan annotation of MS spectra; TAG Expression annotates glycan signals on MALDI-TOF MS spectra based on the glycan list, quantifies the annotated glycans, and provides outputs in a standard format; and TAG Pathway maps glycan expression variation to glycan biosynthetic pathways. The glycan list made with TAG List is defined as semi-comprehensive in the sense that it includes all *N*-glycans and free *N*-glycans in a certain range based on known biosynthetic pathways, but not all glycans can be listed universally. Glycan biosynthetic pathways can be visualized using the KEGG Pathway, Roche Biochemical Pathway, and Reactome databases [9,10,11], but these databases include only a fraction of all representative pathways. The pathways employed in TAG are not also comprehensive, but the TAG approach is modifiable and expandable by users. In addition, we employed a fused pathway in which *N*-glycans and free oligosaccharides (FOSs) are interconnected, and we viewed glycan expression variation data obtained by TAG Expression using this fused pathway. The usefulness of TAG was validated using glycomic expression profile data from Chinese hamster ovary (CHO) cells and mutants lacking the *Npc1* gene (*Npc1* knockout (KO) CHO cells).

## 2. Materials and Methods

### 2.1. Overview of Toolbox Accelerating Glycomics (TAG)

An overview of glycomic analysis using TAG is shown in Figure 1. A multiple-tabbed Excel file containing sets of peak positions and peak area obtained from MALDI-TOF MS data provides the input. In the Excel file, each tab corresponds to a separate measurement, each MS measurement is called an ‘experiment’, and a ‘series’ is defined as a group of experiments with certain conditions, such as wild type, disease, gene KO cell type, etc. For example, nine tabs are obtained for n = 3 experiments with three different series, resulting in nine comma-delimited files (CSVs). The current version of TAG has the functions listed in Table 1. TAG Expression annotates glycans using these CSV files, and glycans are listed by database searching. The TAG List constructs semi-comprehensive glycan lists that are used by TAG Expression. Annotated glycans are quantified based on the amount of added glycan internal standard. The resulting glycan structure and expression level data allow automated statistical analysis approaches such as calculating averages and standard deviations as well as performing Student’s t-tests and hierarchical cluster analysis. The results are outputted as CSV files, and commercially available spreadsheet software can be used to examine the analyzed data. TAG Expression also generates an input file for cluster analysis by Cluster 3.0 software [12,13]. The results of cluster analysis can be viewed using Java Treeview [14]. The TAG Pathway visualizes the observed variation in glycan expression in the results from TAG Expression based on glycan biosynthetic pathways. The TAG Pathway can be used for *N*-glycan, FOS, and combined analyses with *N*-glycan+FOS. With respect to FOS, we focus on the free *N*-glycans (FNGs) in this study.

### 2.2. Implementation of TAG

#### 2.2.1. Interface

Although TAG is essentially a set of program codes that run on a Microsoft Windows platform, some of the programs have been ported to the Mac OSX platform. The TAG interface is written in the Tcl/Tk language. The startup screen, shown in the center of Figure 1, includes various buttons for carrying out expression analysis and generating a glycan list by selecting the files needed for each process. The current version of TAG requires Gnuplot [15], Cluster 3.0, and gawk [16] for graphing, hierarchical cluster analysis, and running the TAG List and TAG Pathway functions, respectively.

TAG Expression is written in FORTRAN 77, and the executable modules were made with GNU Fortran on the Cygwin platform. The TAG List and TAG Pathway are written as awk scripts, TAG List programs are typically run using command lines, and TAG Pathway procedures are controlled via the tcl/tk startup screen.

#### 2.2.2. TAG List

Figure 2 shows the glycan list generated by the TAG List that is used by TAG Expression to automatically assign glycan signals. An ‘N’ or ‘F’ character in the first row indicated whether the list contains *N*-glycan or FNG data. The labels and *m*/*z* values of internal standards are in the second row and separated by commas. The glycan composition, *m*/*z*, and types summarized in Appendix A (*N*-glycans) and Appendix A (FNGs) are below the third row; these are also separated by commas. A typical list for *N*-glycan and FNG data is made using the TAG List program.

*N*-glycans that are mainly composed of mannose (Man), *N*-acetylglucosamine (GlcNAc), *N*-acetylgalactosamine (GalNAc), galactose (Gal), fucose (Fuc), and sialic acid (*N*-acetylneumraminic (NeuAc) acid and *N*-glycolylneuraminic acid (NeuGc) are classified into four groups: high (oligo) mannose type (HM, (Man)5-9(GlcNAc)2); pauci mannose type (PM, (Man)1-4(GlcNAc)2(Fuc)0-1); complex type containing various numbers of Gal, GlcNAc, sialic acid (NeuAc or NeuGc), Fuc, and other residues linked to a (Man)3(GlcNAc)2 *N*-glycan core structure, and hybrid type, which is a hybrid of high mannose and complex oligosaccharide. Mono- to tri-glucosylated HMs ((Glc)1-3(Man)9(GlcNAc)2), which are precursors of HM glycans, also exist. In any case, the core structure consists of three Man residues and two GlcNAc residues ((Man)3(GlcNAc)2). To evaluate the glycan expression profile, it is useful to further classify the structures of *N*-glycans according to the number of branches, neutral or acidic, presence or absence of fucose residues, and other features. For this purpose, we classified *N*-glycans into 43 types as defined in Appendix A. In the case of FNGs, they were classified into 82 (Appendix A). A major difference is that FNGs have two types of core structure: (Man)3 (GlcNAc)2 (N2 core) and (Man)3 (GlcNAc)1 (N1 core). In the current study, we employed high-mannose, pauci-mannose, hybrid, and complex glycans with mono- to tetra-antennae, with up to three Fuc and up to two acetyl modifications, resulting in TAG List files containing 900 *N*-glycans and 1770 FNGs.

Although the structures of complex *N*-glycans are highly diverse because the core structure is modified with various monosaccharides, there are some general rules regarding the structure of complex type *N*-glycans [2]. For example, core mannose is modified with GlcNAc(s), and the number of GlcNAc groups determines the number of branches (typically one to four). GlcNAc is often modified with Gal (and/or GalNAc) to generate LacNAc or LacdiNAc structures, respectively. Fucosylation occurs on the non-reducing end of Gal (H-type), on GlcNAc in branches (lewis type outer arm fucosylation), and on reducing terminal GlcNAc (core fucosylation). Sialylation often occurs on terminal Gal groups with α2-3 and α2-6-linkages. Hybrid *N*-glycans contain mannose residues ((Man)1-3) on one side of the branch, while the other side has a complex-type sugar chain containing GlcNAc modified by α-1,3-mannosyl-glycoprotein 2-β-*N*-acetylglucosaminyltransferase. Using the resulting information, we defined structures of complex and hybrid *N*-glycans to generate a semi-comprehensive *N*-glycan list. In addition to these structures, other modifications such as polylactosamine structure, sulfates, glucuronic acids, and α-2-8-linked polysialic acid structures may also exist, but these will be considered in future versions of the program, and they were not considered in the current work.

To make the annotation of glycans more concise, we employed five-digit notation, in which each digit expresses the number of Hex (Man, Gal, Glc), HexNAc (GlcNAc, GalNAc), Fuc, NeuAc, and NeuGc groups, excluding core structures. In case of *N*-glycans, (Man)3(GlcNAc)2 core structures are shown as ‘C’, while in the case of FNGs (Man)3(GlcNAc)2, core structures are shown as ‘N2′, and (Man)3(GlcNAc)1 core structures are shown as ‘N1′. The *m*/*z* value for each glycan was calculated by assuming that the glycans were derivatized with aoWR (an MS-friendly tag with a high proton affinity), and carboxylic acids of sialic acids were methylesterified. [17] Since the TAG list is a simple awk script, it is easy to modify, allowing users to analyze data obtained using other labeling reagents and modifications.

#### 2.2.3. TAG Expression

CSV input files are extracted from Excel files (xls files) outputted from MS instruments. Since almost all MALDI-TOF MS instruments can export *m*/*z* and signal strength (e.g., height, area) values of spectra signals as xls files, we chose to use the exported xls files as source data for TAG. An example of an input file, a Tab Separated Values file produced by FlexAnalysis 3.0 (Bruker Daltonics, Bremen, Germany), is shown in Figure 3. In this mass-list file, rows 1 to 3 are used as headers, and rows 4 and below contain data obtained by MALDI-TOF MS analysis (e.g., *m*/*z*, time, intensity, etc.). The current version of TAG utilizes *m*/*z* (1st column) and peak area (7th column) values.

Some additional information (metadata) for analysis, such as the quantity of added internal standards (pmol), the error tolerance for precursor *m*/*z* values, and the amount of protein used for analysis (glycan expression is normalized per 100 µg protein), must be entered in the second row of the CSV by the user, as shown in Table 2. TAG Expression utilizes the glycan list generated by TAG List to annotate glycan signals.

Following the process performed by TAG Expression, a number of files are generated in a designated folder. These are CSV sheets of the results, as well as various images for chart building. An ‘out_list.csv’ file stores data related to the annotation of glycans, as described in the results and discussion.

We observed that the amplitude of measurement errors tends to vary depending on *m*/*z* and range values. TAG Expression generates a file named ‘calib_ms_value_plot.html’, and scatter diagrams are drawn (Figure 4) in which the x-axis is *m*/*z* (theoretical) and the y-axis is the deviation between theoretical and observed *m*/*z* values. In the diagram, the maximum cluster and outliers are considered to be signals from glycans and contaminants, respectively (Figure 4). To validate this approach, we confirm that the maximum cluster includes the signal from added internal standards after selection of the cluster.

Based on previous findings that oligosaccharides within a certain molecular weight range may exhibit similar signal strengths irrespective of their structure [18,19], when examined by an appropriately calibrated MALDI-TOF MS instrument, we employed absolute quantification by comparative analysis of the areas of the MS signals derived from each glycan and a known amount of the internal standard (A2GN1). It should be noted that TAG Expression not only annotates glycan signals, but also outputs a table in which assigned glycans are comparably sorted using quantitative information in a standardized format. Specifically, the average and standard deviation for each glycan concentration are calculated automatically, along with t-test analysis of all possible combinations of experimental groups. TAG Expression also produces ‘Exp_list.csv’ files and ‘each_glycan_quant.html’ files that summarize the quantitative expression profiles and plot graphs comparing average glycan expression values with standard deviation between groups, respectively (Appendix A (*N*-glycans) and Appendix A (FNGs)). Additionally, script files for gnuplot chart production and Cluster 3.0 cluster analysis are produced.

#### 2.2.4. TAG Pathway

When a cellular *N*-glycome is analyzed, some tens to several hundreds of glycans may be detected on MALDI-TOF spectra. If glycan expression variation can be mapped against a biosynthetic pathway, an overview of metabolic changes in the system can be obtained.

In eukaryotes, *N*-Glycan synthesis [2] begins with the synthesis of an oligosaccharide containing 14 monosaccharides on the lipid dolichol phosphate, following which the lipid-linked oligosaccharide is transferred to a specific asparagine residue of the growing polypeptide chain by the action of oligosaccharyltransferase (OST) in the endoplasmic reticulum (ER). Correctly folded glycoproteins transit to the Golgi apparatus to form high-mannose, hybrid, and complex *N*-glycans. In mammalian cells, FNGs are generated by three metabolic pathways: (i) OST-mediated release of FNGs in the ER [20,21], (ii) pyrophosphatase acting on dolichol-linked oligosaccharides [22,23], and (iii) cytoplasmic peptide:*N*-glycanase (PNGase) acting on misfolded glycoproteins via ER-associated degradation (ERAD) [24,25]. In any case, endo-β-*N*-acetylglucosaminidase (ENGase) in the cytosol metabolizes FNGs to Gn1 glycans, which have only a single GlcNAc at their reducing termini. These Gn1 glycans are susceptible to the action of a cytosolic α-mannosidase (Man2C1), giving rise to the specific Man5 GlcNAc1 structure. Man5 GlcNAc1 is transported into lysosomes where the oligosaccharides are hydrolyzed into monosaccharides for recycling. The TAG Pathway draws a fused biosynthesis pathway for *N*-glycans and FNGs; then, the variation in glycan expression is mapped onto this fused pathway.

The TAG Pathway generates an html file of glycan biosynthetic pathways. We prepared biosynthetic pathway maps for *N*-glycans and FNGs in CSV format, part of which is shown in Figure 5. The map includes cells in an Excel sheet containing information on glycans (e.g., from cells BS39 to BS42) such as glycan class (in BS39, *N*-glycans on folded proteins), type of glycan (in BS40, HM means high mannose-type glycans), five-digit notation of glycan composition (BS41), and localization of glycans (in BS42, Golgi and later) or relationships between glycans (e.g., ‘ra.png’ means right arrow image in cell BR38). The image files are located in the ‘glycan_img’ folder of TAG. The intracellular localization of glycans is expressed using cell color, where yellow, pale blue, gray, and magenta indicate ER, Golgi, cytoplasm, and lysosome, respectively. These files can be easily written or modified by users.

### 2.3. Experimental Glycomic Analysis of N-glycans and FNGs for Npc1 KO CHO Cells

To evaluate the feasibility of TAG, *N*- and FNG-glycomic data from our previously reported CHO cells and mutants lacking a functional *Npc1* gene (*Npc1* KO CHO cells), herein denoted NPC(−), were used as a test case [26]. The *Npc1* gene encodes a protein involved in lipid transport between lysosomes and the ER. The *Npc1* KO CHO cells provide a model for Niemann–Pick disease type C (NPC) [26]. *Npc1* deficiency causes an accumulation of free cholesterol and glycolipids, and 2-Hydroxypropyl-β-cyclodextrin (HPBCD) can reduce cholesterol accumulation. Our previous study examined the effects of *Npc1* KO and HPBCD treatment on glycome expression. Glycomic analyses were carried out on wild-type CHO cells (denoted as wt(−)), *Npc1* KO CHO cells (denoted as NPC(−), and *Npc1* KO CHO cells treated with HPBCD (denoted as NPC(+)) using TAG, where the (+) and (−) symbols indicate whether HPBCD was added or not. The results were compared with those from our previous study using a macro and Microsoft Excel.

#### 2.3.1. Extraction of Cellular Glycoproteins and Free Oligosaccharides

For the *N*-glycan and FNG analyses, glycoproteins and FNGs were extracted as previously described [17]. Approximately 1 × 10^6^ cells were suspended in Tris-acetate buffer containing 2% sodium dodecyl sulfate and homogenized using an Ultrasonic Homogenizer (Taitec, Saitama, Japan). Reductive alkylation of the cellular proteins was performed, followed by the precipitation of proteins in the presence of a four-fold volume of ice-cold ethanol. The precipitates including proteins were dried, dissolved in ammonium bicarbonate, and digested with trypsin. Finally, *N*-glycans were prepared by PNGase F difestion. The supernatants containing FNG were completely dried and dissolved in deionized water. Then, the samples were directly subjected to the glycoblotting procedure. Detailed procedures and are provided elsewhere [26].

#### 2.3.2. Glycoblotting Procedure and MALDI-TOF/TOF MS Analysis

*N*-glycans and FNGs were subjected to the glycoblotting procedure. Detailed procedures and materials are provided elsewhere [26,27]. Purified *N*-glycans, FNGs, were combined with 2,5-dihydrobenzoic acid (10 mg/mL in30% acetonitrile) and subsequently subjected to MALDI-TOF MS analysis, as previously described [17]. As all MALDI-TOF MS instruments may have a function to export the values of *m*/*z* and signal strength (e.g., height, area) of observed signals on the spectra as xls files, we chose to use the exported xls files as a source data of TAG.

## 3. Results and Discussion

The usefulness of TAG was explored using our previously reported *Npc1* KO CHO cell data [26].

### 3.1. Glycan Expression Analysis of Npc1 KO CHO Cells

The ‘out_list.csv’ file is shown in Figure 6. For signals with *m*/*z* values within the range of error tolerance, information about the composition, five-digit notation, glycan type as defined in Appendix A, theoretical *m*/*z* deviation between *m*/*z* (theoretical) and *m*/*z* (observed), area, and determined quantity (pmol/100 µg protein) are shown. Values of glycans not detected in the analysis are shown as zero.

As shown in Figure 7, the ‘exp_list.csv’ file summarizes the quantitative expression profiles in terms of the quantity of each glycan (pmol/100 µg protein), the average, standard deviation (S.D.), and coefficient of variation (C.V.) of defined groups, and t-values and p-values (Student’s t-test) between all possible combinations of series. In addition to Student’s t-tests, hierarchical clustering was applied in order to group glycans based on their expression levels among samples. This was performed using the open source tool Cluster 3.0 to create clusters of data, and these clusters were represented graphically using the open source tool Java TreeView [14].

The number of unique glycans identified by TAG in the present work was compared with that reported in our previous study [26]. Since TAG annotates all glycans with the same *m*/*z* values but different structures, we collected only *m*/*z* values to represent the unique glycans annotated. *N*-glycans and FNGs detected and quantified by TAG Expression are summarized in Appendix A. A total of 61 and 36 glycans with unique *m/z* were assigned as *N*-glycans and FNGs, respectively, by TAG Expression. These numbers are greater than those previously reported (58 *N*-glycans and 30 FNGs) based on the manual picking of glycan signals [26]. It should be noted that all *N*-glycans and FNGs detected in our previous work [26] were successfully detected in the present study. Absolute quantities of glycans measured in both the previous and current study are in good agreement, with correlation coefficients (r^2^) > 0.999. In addition, we generated glycans containing 0–2 acetyl groups for *N*-glycans and FNGs using the TAG List, resulting in two acetylated *N*-glycans and an acetylated FNG that was not detected in our previous study. One *N*-glycan and five FNG signals were also annotated in the TAG results that were similarly not detected in our previous study. This indicates the importance of the semi-comprehensive glycan list feature of TAG List. Thus, the modifiable and expandable nature of the TAG List function provides great advantages for correctly and comprehensively detecting glycan signals without the need for detailed knowledge on glycan structure.

### 3.2. Glycan Expression Variation Mapped to Biosynthetic Pathways

The biosynthetic pathway used for the analysis of *N*-glycans and FNGs is shown in Figure 8. Each cell (small rectangle) corresponds to an individual glycan. *N*-glycans are shown to be attached to mature, immature, or misfolded proteins. The background color of the cell indicates the localization of glycans (e.g., cytoplasm, ER, Golgi, lysosome). The relevance of the colors is described above. The TAG Pathway creates two types of biosynthetic pathway maps; one shows variation in glycan expression as bar charts, and the other generates numerical values such as expression differences and ratios. In the latter case, significant increases and decreases (*p* < 0.05) are highlighted on the pathway in red and blue, respectively. Figure 9 shows the data in each cell, which include glycan structure, five-digit notation, and a bar chart showing glycan expression (Figure 9 left). Rather than a bar chart, we can visualize expression variability as differences, ratios, or *p*-values (Figure 9 right).

Several pioneering studies reported various glycomic alterations in NPC involving various glycosphingolipids (GSL) (glucosylceramides (GlcCer), lactosylceramide (LacCer), GM2, GM3, and asiao-GM2) [28,29]. In addition, the disease-specific accumulation of various sialylated glycoconjugates within endocytic compartments of *Npc1*-null and *Npc2*-deficient fibroblasts is caused by impaired recycling as opposed to altered fusion of vesicles. The treatment of either *Npc1*-null or *Npc2*-deficient cells with cyclodextrin was effective in reducing cholesterol storage, as well as the endocytic accumulation of sialylated glycoconjugates [30], although structure-intensive analysis was not performed in this study. Our previous study using *Npc1* KO CHO cells identified a number of glycomic alterations, including an increased expression of LacCer, GM1, GM2, GD1, various neolacto-series glycosphingolipids, and sialyl-T (*O*-glycan), as well as various *N*-glycans, which were typically both fucosylated and sialylated. We also observed significant increases in the total amounts of free *N*-glycans (FNGs), especially in the unique complex- and hybrid-type FNGs. The treatment of *Npc1* KO CHO cells with 2-hydroxypropyl-β-cyclodextrin (HPBCD) did not affect the glycomic alterations observed in the GSL-, *N*-, and *O*-glycans of *Npc1* KO CHO cells. However, HPBCD treatment corrected the glycomic alterations observed in FNGs to levels observed in wild-type cells [26].

Glycomic alteration among wt(−), NPC(−), and NPC(+) series can be shown graphically on the map as exemplified in Figure 10. The whole graphic can be seen in Appendix A. In Figure 10a, representative glycomic alterations of HM and PM glycans were shown with bar charts (upper) and mean relative abundance and *p*-value (lower). It is shown that *Npc1* knockout generally increases the glycan expression levels and that HPBCD addition has little effect on glycan expression levels. As shown in Figure 10b–d, fucosylated and sialylated complex type *N*-glycans tend to increase by *Npc1* knockout, some of which increase significantly, and the treatment of *Npc1* KO CHO cells with HPBCD did not often affect the glycomic alterations of *Npc1* KO CHO cells. In contrast, the glycomic alterations observed in FNGs (typically complex type FNGs) were corrected by the treatment of HBPCD to levels observed in wild-type cells. These observations agree well with those reported in our previous publication [26].

These functions will help researchers readily determine the specificity of alterations. Another feature of the TAG pathway is that the map template itself is written in CSV format, and it is modifiable by the user.

## 4. Conclusions

In this study, we developed Toolbox Accelerating Glycomics (TAG) software that provides a modifiable and expandable glycan list, automatically annotates and quantifies the glycan signals on MALDI-TOF MS spectra, quantifies and sorts the results, and visualizes variation in glycan expression based on glycan biosynthetic pathways. Glycomic analysis without such tools requires much time and labor, whereas TAG analysis takes just a few minutes. The results can be viewed by web browsers such as Firefox, Chrome, Edge, and Safari, which reduces instrument dependency. To support the discrimination of structural isomers, TAG can provide a list of possible structure candidates with the same composition, which assists the design of further experiments such as MS/MS and exoglycosidase digestion. Considering that glycomic alterations may be mutually related to one another through biosynthetic pathways, mapping glycomic alterations against biosynthetic pathways can help determine glycan structures. There are a variety of methods to label the reducing termini of glycans and to derivatize sialic acid residue(s). Examples of the former include 2-aminobenzamide [31] and RapiFluor-MS [32]. As examples of the latter, we recently reported a sialic acid linkage-specific alkylamidation method by lactone ring-opening aminolysis (aminolysis-SALSA), for the discrimination between α2,3- and α2,6-linked sialylated glycan isomers by mass spectrometry [33,34]. The TAG List can be readily applied to such labels and derivatizations with minor modification. Although it is desirable to perform additional experiments to distinguish structural isomers, we think it is worth attempting to predict the correct structural isomer by accumulating glycomic data and utilizing machine learning techniques. TAG is a flexible program, and glycan lists and pathway maps can be generated by the researchers themselves. Therefore, we believe that TAG could accelerate glycomic analysis and assist the development of similar software. The TAG programs will be available to all researchers. It will be released via an open source repository service such as GitHub. In addition, we also plan to expand the TAG procedure for the analysis of other glycans, such as glycans of glycosphingolipids, *O*-glycans, and glycosaminoglycans.

## Figures and Tables

**Figure 1 biomolecules-10-01383-f001:**
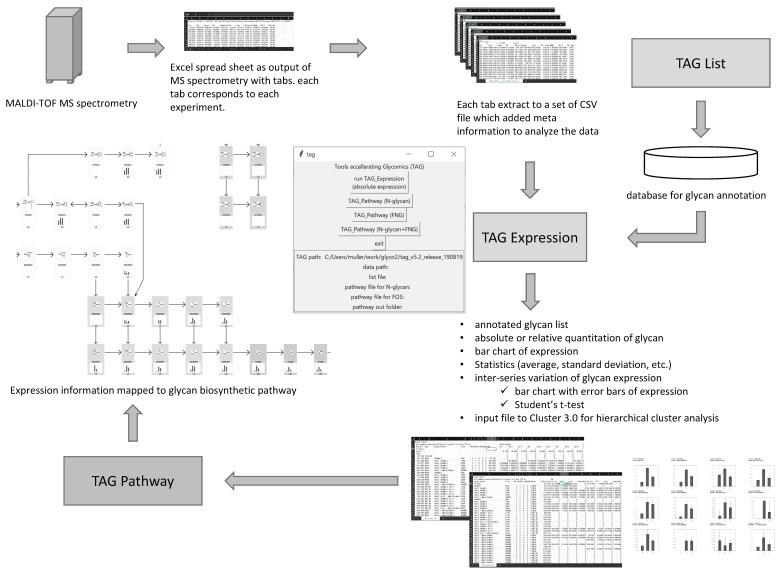
Overview of Toolbox Accelerating Glycomics (TAG).

**Figure 2 biomolecules-10-01383-f002:**
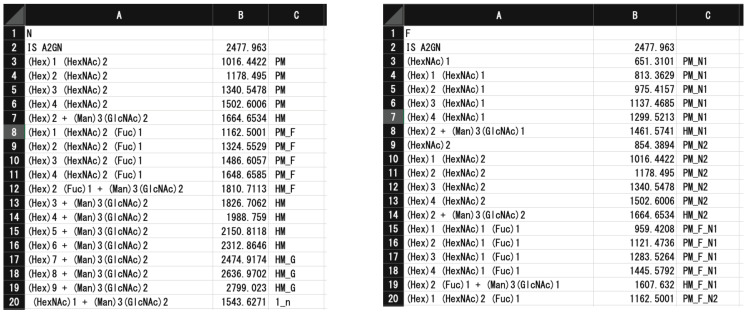
Examples of glycan lists (*N*-glycans on the left, free *N*-glycans (FNGs) on the right) generated by TAG List.

**Figure 3 biomolecules-10-01383-f003:**
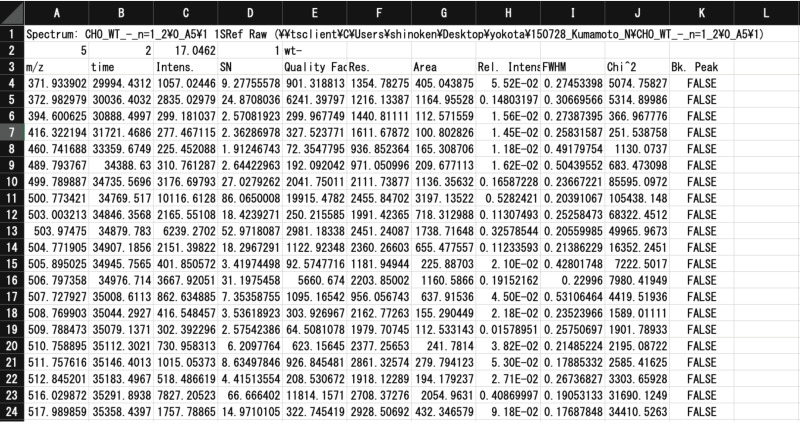
Input file for TAG Expression, a Tab Separated Values file produced by FlexAnalysis 3.0.

**Figure 4 biomolecules-10-01383-f004:**
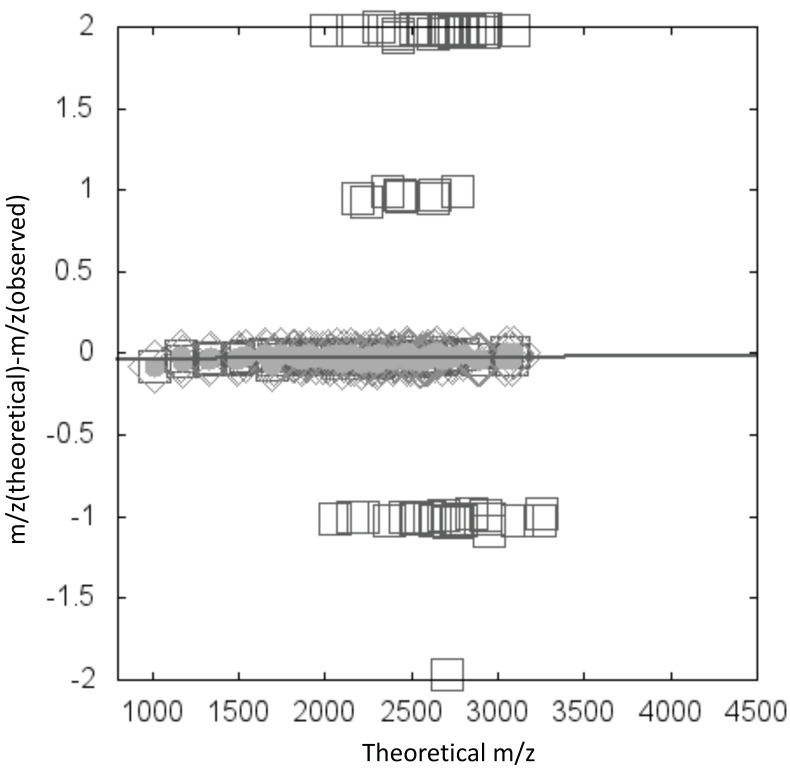
Example of a scatter diagram in which the x-axis is *m*/*z* (theoretical) and the y-axis is the deviation between theoretical and observed *m*/*z* values. Diamonds indicate the maximum cluster. Data are fitted to a linear function, and line indicates the fit. Gray circles indicate annotated glycan data points within a certain distance from the fitted line.

**Figure 5 biomolecules-10-01383-f005:**
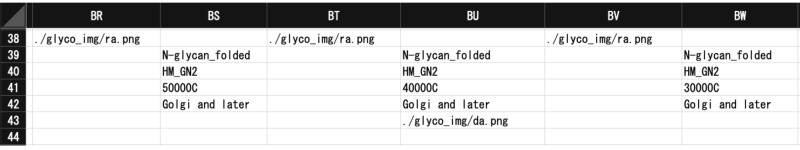
Part of the biosynthesis pathway map in comma-delimited files (CSV) format.

**Figure 6 biomolecules-10-01383-f006:**
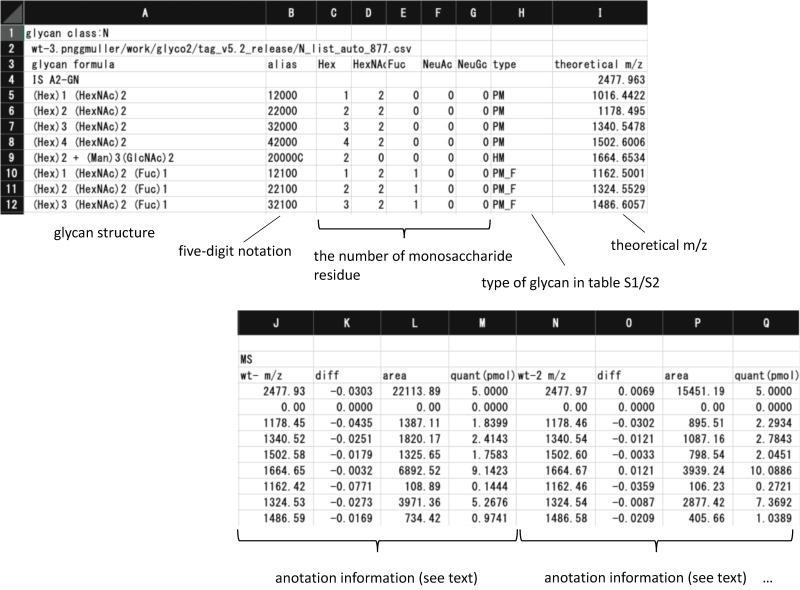
Part of the ‘out_list.csv’ output file for glycan annotation by TAG Expression.

**Figure 7 biomolecules-10-01383-f007:**
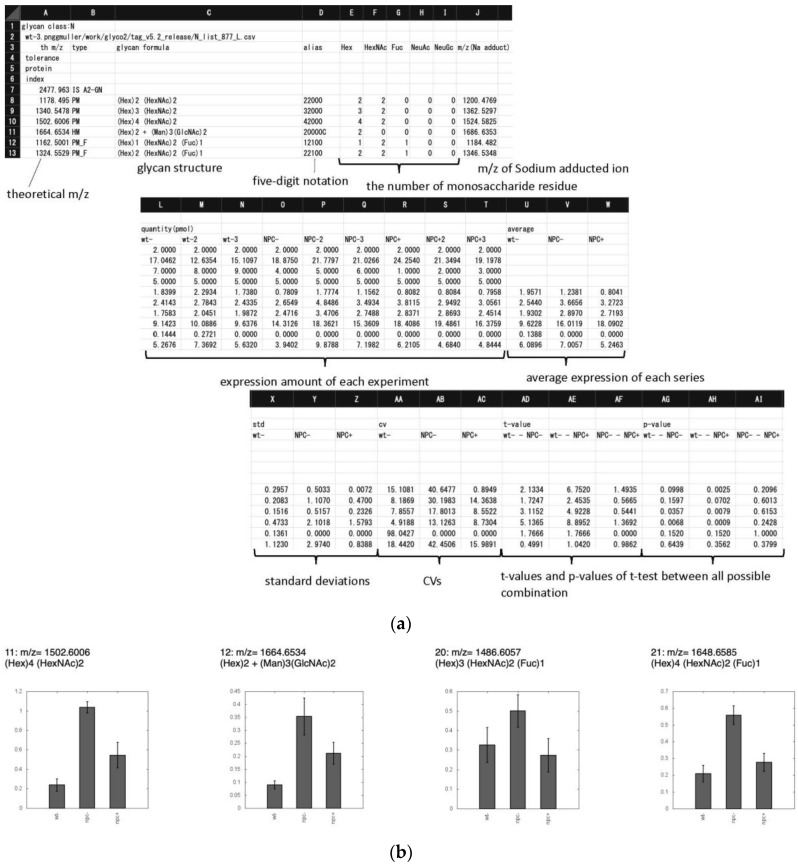
Examples showing the output files (**a**) expt_list.csv that summarize quantitative data and statistics, and (**b**) each_glycan_quant.html that graphs quantitative data for different groups. In each chart, *m*/*z* values and glycan composition are in the header. The vertical axis shows expression quantities (pmol/100 µg protein). Error bars represent standard deviation. Abscissa axis indicate wt(−), NPC(−), and NPC(+), respectively.

**Figure 8 biomolecules-10-01383-f008:**
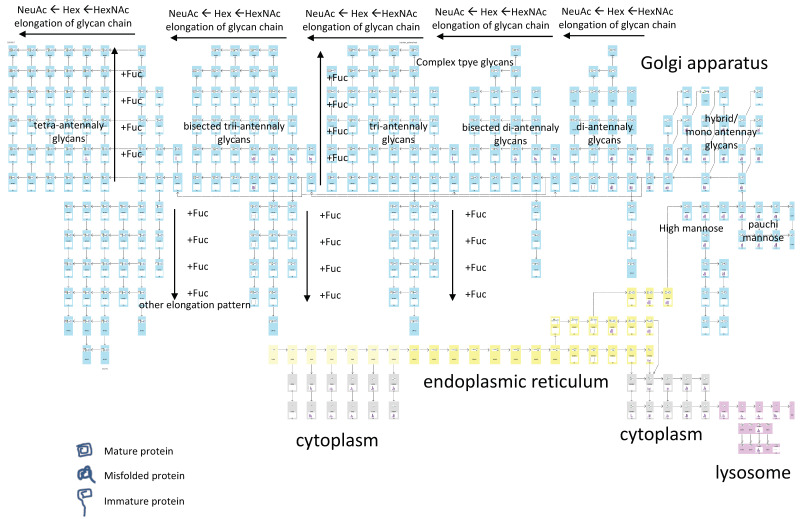
Overview of the structure for the biosynthetic pathway model for *N*-glycans and FNGs.

**Figure 9 biomolecules-10-01383-f009:**
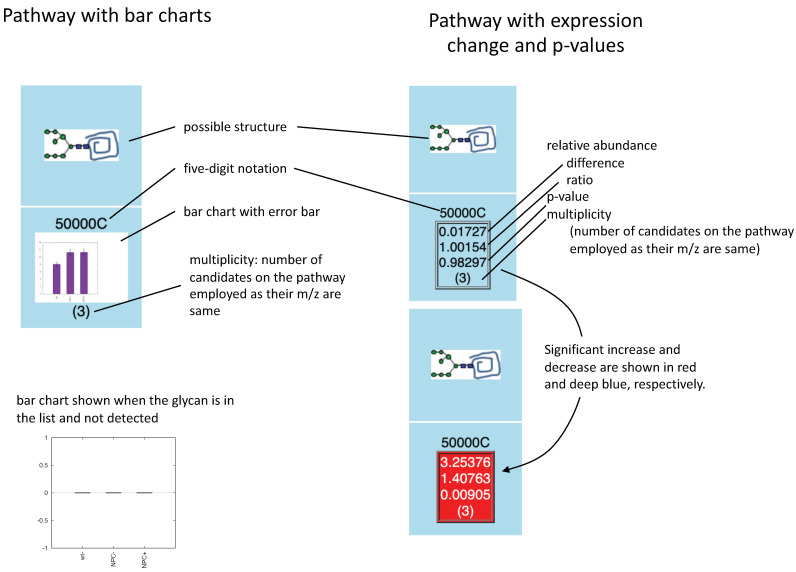
Two different methods for visualizing alterations; bar chart (**left**) and mean relative abundance and *p*-value (**right**). In the bar chart, the three bars show wt(−), NPC(−), and NPC(+) expression levels, respectively.

**Figure 10 biomolecules-10-01383-f010:**
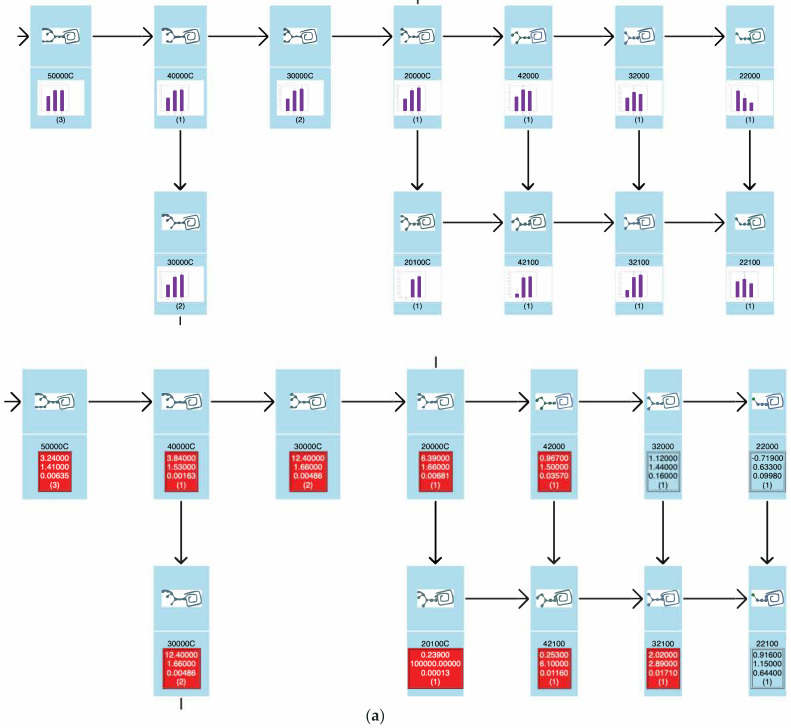
Representative alterations observed among wt(−), NPC(−), and NPC(+) series using bar charts (upper in each panel) and with mean relative abundance and *p*-value between wt(−) and NPC(−) (lower) for *N*-glycans and FNGs. (**a**) Expression levels of high (oligo) mannose type (HM) and pauci mannose type (PM) *N*-glycans. (**b**) Complex biantennary *N*-glycans. (**c**) Complex triantennary *N*-glycans. (**d**) Complex tetraantennary *N*-glycans. (**e**) FNGs possibly derived through the action of pyrophosphatase. (**f**) FNGs possibly derived through the actions of oligosaccharyltransferase (OST) and/or peptide:*N*-glycanase (PNGase). (**g**) for a part of complex FNGs.

**Table 1 biomolecules-10-01383-t001:** Functions of Toolbox Accelerating Glycomics (TAG).

Button Name	Function
TAG List	Generating a glycan list to be employed in TAG Expression. TAG List is a separate program written in the awk script language.
TAG Expression	Glycan annotation, quantitation, statistical analysis, graphing.
TAG Pathway (*N*-glycan)	Visualizing variation in *N*-glycan expression based on *N*-glycan biosynthetic pathways.
TAG Pathway (FOS)	Visualizing variation in FNG expression based on FNG biosynthetic pathways.
TAG Pathway (*N*-glycan + FOS)	Visualizing variation in both *N*-glycans and FNGs based on *N*-glycan and FNG biosynthetic pathways.

**Table 2 biomolecules-10-01383-t002:** Metadata for Glycan Analysis Using TAG Expression.

Cell Position	Data Type	Data
A2	Real number	Quantity of internal standard.
B2	Real number	Error tolerance for precursor *m*/*z*.
C2	Real number	Total quantity (µg) of protein.
D2	Integer	Number to distinguish experimental groups. For example, in the case of the current analysis (see text), wt(−) ^1^ is 1, NPC(−) is 2, and NPC(+) ^1^ is 3.
E2	Strings	Short name of the experiment. For example, in the case of the current analysis (see text), wt(−), NPC(−), and NPC(+) fall under this category.

^1^ The (+) and (−) symbols indicate whether HPBCD was added or not.

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
