# Peer review of "Toolbox Accelerating Glycomics (TAG): Glycan Annotation from MALDI-TOF MS Spectra and Mapping Expression Variation to Biosynthetic Pathways"

_biomolecules, 2020, doi:10.3390/biom10101383_

Round 1
Reviewer 1 Report
- The authors should clarify what is meant by 'semi-comprehensive glycan list' or 'relatively comprehensive'. Does this refer to the possibility that additional N-glycans could be identified or that O-glycans and glycosphingolipids are not included?
- In Materials and Methods, please define FOS. This presumably stands for free oligosaccaharide.
- Is the TAG program available to other investigators? Use by other groups will be important for validation of its usefulness.
- I think the biggest weakness of the manuscript relates to the figures. The images of the spreadsheets are often difficult to read, particularly those in Figures 6 and 7. Similarly, the images in Figures 9 and 10 are so small, that little can be gained from them. Figure 8 can be removed as it does not add any further information beyond what is described in the text.
- There are a few typos in the figures, chrat rather than chart, tehoretical rather than theoretical.
Author Response
Response to Reviewer 1 Comments
Porint 1: The authors should clarify what is meant by 'semi-comprehensive glycan list' or 'relatively comprehensive'. Does this refer to the possibility that additional N-glycans could be identified or that O-glycans and glycosphingolipids are not included?
Response 1: We strongly appreciate the reviewer's comment on this point. As the reviewer pointed out, 'semi-comprehensive' and 'relatively comprehensive' were confusing statements. We use the term ‘semi-comprehensive’ in the sense that it includes all N-glycans and free N-glycans in a certain range based on known biosynthetic pathways, but not all glycans can be listed universally, therefore, we added the definition on page 2 line 64 in the revised text as follows.
“The glycan list made with TAG List is defined as semi-comprehensive in the sense that it includes all N-glycans and free N-glycans in a certain range based on known biosynthetic pathways, but not all glycans can be listed universally.”
Since the mean of “relatively comprehensive” in our article is the same as “semi-comprehensive”, the “relatively comprehensive” at the fourth line from the end of the third paragraph of 2.2.1(line 182)of revised manuscript was changed to “semi-comprehensive”.
Point 2: In Materials and Methods, please define FOS. This presumably stands for free oligosaccaharide.
Response 2: We appreciate the reviewer's comment on this point. Definition of FOSs is shown in the Introduction (line 70), where term FOS first appears.
Point 3: Is the TAG program available to other investigators? Use by other groups will be important for validation of its usefulness.
Response 3: We appreciate the reviewer's comment on this point. TAG program will be available to other researchers. As soon as the packaging of the program is ready, we plan to release it to the world via a repository service such as GitHub. We modified a sentence in the second sentence from the end of the Conclusions (line 469) as follows.
“The TAG programs will be available all researchers. It will be released via open source repository service such as GitHub.”
Point 4: I think the biggest weakness of the manuscript relates to the figures. The images of the spreadsheets are often difficult to read, particularly those in Figures 6 and 7. Similarly, the images in Figures 9 and 10 are so small, that little can be gained from them. Figure 8 can be removed as it does not add any further information beyond what is described in the text.
Response 4: For Figure 6 and 7, we have split the captured Excel sheet into smaller images in order to make the figure more readable. Figure 8 is removed, according to the reviewer’s comment and subsequent figures have been renumbered accordingly. As pointed out by the reviewer, Figure 9 (in the revised manuscript Figure8) is now further enlarged which is rotated 90 degrees to make the image as large as possible. According to the suggestion of the reviewer, we have separated Figure 10 into two figures (Figure 9 and Figure 10 in the revised manuscript), to make the chart more readable. In addition, according to the reviewer’s comment, we have adjusted the size of the diagrams to allow readers see as much detail as possible. Furthermore, after submitting the manuscript, we found some errors in Figure S3-S5 and we fixed those. Figure 10 is based on the revised Figures. The Figure caption of the Figure 9 and 10 were also revised as follows.
“Figure 9. Two different methods for visualizing alterations; bar chart (left) and mean relative abundance and p-value (right). In the bar chart, the three bars show wt(-), NPC(-) and NPC(+) expression levels, respectively.”
“Figure 10. Representative alterations observed among wt(-), NPC(-), and NPC(+) series using bar charts (upper in each panel) and with mean relative abundance and p-value between wt(-) and NPC(-) (lower) for N-glycans and FNGs; (a) expression levels of HM and PM N-glycans. (b) complex biantennary N-glycans (c) complex triantennary N-glycans (d) complex tetraantennary N-glycans. (e) FNGs possibly derived through the action of pyrophosphatase. (f) FNGs possibly derived through the actions of OST and/or PNGase. (g) complex FNGs.”
We also indicated that the whole picture can be viewed through Figure S3-S5 in the first line of the third paragraph of the subsection 3.2. Other figures such as Figure 2, 3, and 5 were also enlarged as much as possible.
In accordance with removing the Figure 8, the first line in the third paragraph of subsection 3.1 (line 338) is modified as follows.
“The number of unique glycans identified by TAG in the present work was compared with that reported in our previous study [26].”
The first sentence in the subsection 3.2 (line 359) were modified as follows in accordance with the numbering modification from figure 9 to figure 8.
“The biosynthetic pathway used for the analysis of N-glycans and FNGs is shown in Figure 8.”
In accordance with the modification of Figure 10, the four lines at the end of the third paragraph of the 3.2 (line 366) is revised as follows.
“Figure 9 shows the data in each cell, which includes glycan structure, five-digit notation, and a bar chart showing glycan expression (Figure 9 left). Rather than a bar chart, we can visualize expression variability as differences, ratios, or p-values (Figure 9 right).”
Point 5: There are a few typos in the figures, chrat rather than chart, tehoretical rather than theoretical.
Response 5: We appreciate the reviewer's comment on this point. According to reviewer’s comment, the typos in the figure 6, figure 7, and figure 10a (Figure 9 in the revised manuscript) were corrected.
Reviewer 2 Report
In this manuscript entitled “Toolbox accelerating glycomics (TAG): glycan annotation from MALDI-TOF MS spectra and mapping expression variation to biosynthetic pathways”, Miura et al. developed a new software program, called Toolbox accelerating glycomics (TAG), for glycomics analysis on N-glycans and free N-glycans (FNGs) with tremendous heterogeneity. The TAG program contains three elements: the first “TAG List” generates a list of glycans to be searched for, the second “TAG Expression” quantifies and graphs the amount of each glycan in the sample, and the last “TAG Pathway” allows the data obtained to be applied to the biosynthetic pathway. The authors utilized the TAG program for the analysis of the samples from wild type Chinese hamster ovary (CHO) cells and an Npc1-lacking mutant version of CHO cells. The TAG-based analysis resulted in the detection and quantification of 61 of N-glycans and 36 FNGs. These numbers were greater than 58 of N-glycans and 30 FNGs in the previous study which was done manually by the same group. Collectively, the authors concluded that the TAG program will help users not only reduce the amount of time and labor for glycomics analysis but also analyze more types of glycans than with existing methods.
The manuscript is concisely written. The authors explained well about the TAG program. This new program that the authors developed would contribute to the advancement of glycobiology. The manuscript will be appreciated by many readers in the field of glycobiology. Thus, I would like to recommend the publication of this manuscript in Biomolecules after the authors address my concerns below.
(1) I would like the authors to add some information about sample preparation for MALDI-TOF MS in the Materials and Methods section. The main text says that the TAG uses data from MALDI-TOF MS. This does not tell readers how they should (and should not) prepare their samples and analyze them with MALDI-TOF MS at all.
(2) The authors should describe a more detailed analysis of the results in the Results and Discussion section to make a convincing claim about the usefulness of the TAG program. To begin with, what changes in N-glycosylation and FNGs were expected to occur as a result of the genetic loss of Npc1? At a minimum, the information should be provided in this manuscript although I understand that there may be some overlap with Ref. 26.
Figure 10 (c) shows only part of the results. How do the authors interpret these results considering the expectations above? Figure S4 shows that other glycans were also altered. In particular, it is interesting to note that the changes in expression levels are scattered across multiple pathways, rather than all in one pathway. Those should also be discussed.
The authors state that a comparison between NPC(-) and NPC(+) showed no significant changes. However, some glycans showed significant decreases as shown in Figure S5. The authors should explain this discrepancy. Also, the authors should add expectations in this experiment with 2-hydroxypropyl beta-cyclodextrin and discussion about the results in comparison with the expectations.
Also, the authors should justify showing the comparison between wt (-) and NPC (+) in Figure S6 from the biological point of view. To me, the fair comparison at least needs a dataset of wt (+).
(3) In Table 1, the “TAG List” should be listed first. Currently, it is at the bottom of the list. In the sixth sentence (Line 80) in the first paragraph in 2.1 Overview of Toolbox Accelerating Glycomics (TAG), the authors mention about TAG List at first and then TAG List in the next sentence. The authors should change the order.
(4) The eighth sentence (Line 83) in the first paragraph in 2.1 Overview of Toolbox Accelerating Glycomics (TAG) needs a period.
(5) In the fourteenth sentence (Line 90) in the first paragraph in 2.1 Overview of Toolbox Accelerating Glycomics (TAG), FOS should be spelled out.
(6) In the third sentence (Line 121) in the second paragraph in 2.2.2. TAG List, the authors spelled out GlcNAc for the first time although they utilized it earlier in the first sentence in the same paragraph. The authors should fix this issue.
(7) In the second sentence (Line 261) in the third paragraph in 3.1. Glycan expression analysis of Npc1 KO CHO cells, “the of unique glycans annotated.” does not make sense. The authors should fix this issue.
(8) In the seventh and eighth sentences (Lines 267 and 268) in the third paragraph in 3.1. Glycan expression analysis of Npc1 KO CHO cells, some letters were italicized oddly. The authors should fix this issue.
(9) In the eleventh sentence (Lines 272) in the third paragraph in 3.1. Glycan expression analysis of Npc1 KO CHO cells, “provide” should be “provides”.
(10) In Figure 8, there is a reference No. 24. That should be No. 26 instead.
(11) In the figure legend to Figure 10, there is the word “lycomic”. Did the authors mean “glycomic”?
Author Response
Response to Reviewer 2 Comments
Point 1: I would like the authors to add some information about sample preparation for MALDI-TOF MS in the Materials and Methods section. The main text says that the TAG uses data from MALDI-TOF MS. This does not tell readers how they should (and should not) prepare their samples and analyze them with MALDI-TOF MS at all.
Response 1: We strongly appreciate the reviewer's comment on this point. As the reviewer pointed out, the information about the more detailed experiment of glycomics analysis is essential and useful to the readers. We added two subsubsections to provide brief description for sample preparation and MALDI-TOF MS as subsubsections 2.3.1 and 2.3.2 in Materials and methods (line 296-313). In addition, one reference was added.
Point 2-1: The authors should describe a more detailed analysis of the results in the Results and Discussion section to make a convincing claim about the usefulness of the TAG program. To begin with, what changes in N-glycosylation and FNGs were expected to occur as a result of the genetic loss of Npc1? At a minimum, the information should be provided in this manuscript although I understand that there may be some overlap with Ref. 26.
Response 2-1: We wish to express our deep appreciation to the reviewer for his insightful comment on this point. We added following sentences after the first paragraph in subsection 3.2 Glycan expression variation mapped to biosynthetic pathways (line 407) to provide the summary of glycomic alterations in NPC previously reported by other researchers. We also added the summary of glycomic alterations observed in our previous publication using Npc1 KO CHO cells. We added three articles to reference list.
“Several pioneering studies reported various glycomic alterations in NPC involving various glycosphingolipids (GSL) (glucosylceramides (GlcCer), lactosylceramide (LacCer), GM2, GM3, and asiao-GM2) [28,29]. In addition, the disease-specific accumulation of various sialylated glycoconjugates within endocytic compartments of Npc1-null and Npc2-deficient fibroblasts is caused by impaired recycling as opposed to altered fusion of vesicles. Treatment of either NPC1-null or Npc2-deficient cells with cyclodextrin was effective in reducing cholesterol storage, as well as the endocytic accumulation of sialylated glycoconjugates [30], although structure intensive analysis was not performed in this study. Our previous study using Npc1 KO CHO cells identified a number of glycomic alterations, including increased expression of LacCer, GM1, GM2, GD1, various neolacto-series glycosphingolipids, and sialyl-T (O-glycan), as well as various N-glycans, which were typically both fucosylated and sialylated. We also observed significant increases in the total amounts of free N-glycans (FNGs), especially in the unique complex- and hybrid-types FNGs. Treatment of Npc1 KO CHO cells with 2-hydroxypropyl-b-cyclodextrin (HPBCD) did not affect the glycomic alterations observed in the GSL-, N- and O-glycans of Npc1 KO CHO cells. However, HPBCD treatment corrected the glycomic alterations observed in FNGs to levels observed in wild-type cells [26].”
Point 2-2: Figure 10 (c) shows only part of the results. How do the authors interpret these results considering the expectations above? Figure S4 shows that other glycans were also altered. In particular, it is interesting to note that the changes in expression levels are scattered across multiple pathways, rather than all in one pathway. Those should also be discussed.
Response 2-2: As suggested by the reviewer, we added more results in Figure 10 to show representative expression alterations observed in multiple pathways. This addition will help readers to confirm what we summarized about the glycomic alterations observed in our previous publication using Npc1 KO CHO cells (please see the “Response to 2.1”). In addition, after submitting the manuscript, we found some errors in Figure S3-S5 and we fixed those. The revised Figure 10 is based on the revised Figures.
Regarding N-glycans, expression alterations observed for HM and PM, complex biantennay, complex triantennary, and complex tetraantennary glycans are shown in Figure 10 a, b, c, and d, respectively. Expression alterations observed for FNGs possibly derived through the action of pyrophosphatase, possibly derived through the actions of OST and/or PNGase, are shown in Figure 10 e and f, respectively. Expression alterations observed for complex type FNGs, of which biosynthetic pathway is not fully understood, is shown in Figure 10 g.
Point 2-3: The authors state that a comparison between NPC(-) and NPC(+) showed no significant changes. However, some glycans showed significant decreases as shown in Figure S5. The authors should explain this discrepancy. Also, the authors should add expectations in this experiment with 2-hydroxypropyl beta-cyclodextrin and discussion about the results in comparison with the expectations.
Response 2-3: As described in “Response 2-2”, we added more results in Figure 10 to show representative expression alterations observed in multiple pathways. Therefore, the descriptions of the third paragraph of 3.2 (line 422) were also modified as follows.
“Glycomic alteration among wt(-), NPC(-), and NPC(+) series can be shown graphically on the map as exemplified in Figure 10. The whole graphic can be seen in Figure S3-S5. In Figure 10a, representative glycomic alterations of HM and PM glycans were shown with bar charts (upper) and mean relative abundance and p-value (lower). It is shown that Npc1 knock out generally increase the glycan expression levels and that HPBCD addition has little effect on glycan expression levels. As shown in Figure 10b-d, fucosylated and sialylated complex type N-glycans tend to increase by Npc1 knock out, some of which increase significantly, and treatment of Npc1 KO CHO cells with HPBCD did not often affect the glycomic alterations of Npc1 KO CHO cells. In contrast, glycomic alterations observed in FNGs (typically complex type FNGs) were corrected by the treatment of HBPCD to levels observed in wild-type cells. These observations agree well with those reported in our previous publication [26].”
Point 2-4: Also, the authors should justify showing the comparison between wt (-) and NPC (+) in Figure S6 from the biological point of view. To me, the fair comparison at least needs a dataset of wt (+).
Response 2-4: We agree to the reviewer’s comment that comparison should be done also between wt (-) and wt (+). At the moment, however, we think that it is a future task. Therefore, the Figure S6 were removed.
Point 3: In Table 1, the “TAG List” should be listed first. Currently, it is at the bottom of the list. In the sixth sentence (Line 80) in the first paragraph in 2.1 Overview of Toolbox Accelerating Glycomics (TAG), the authors mention about TAG List at first and then TAG List in the next sentence. The authors should change the order.
Response 3: We appreciate the reviewer’s comment. According to the comment, we reordered the rows in Table1 and the “TAG List” was moved to the top raw of the table.
Point 4: The eighth sentence (Line 83) in the first paragraph in 2.1 Overview of Toolbox Accelerating Glycomics (TAG) needs a period.
Response 4: We appreciate the reviewer’s comment. We added a period to the sentence in line 85 of the revised manuscript.
Point 5: In the fourteenth sentence (Line 90) in the first paragraph in 2.1 Overview of Toolbox Accelerating Glycomics (TAG), FOS should be spelled out.
Response 5: We appreciate the reviewer’s comment. We added a definition of the FOS in line 70.
Point 6: In the third sentence (Line 121) in the second paragraph in 2.2.2. TAG List, the authors spelled out GlcNAc for the first time although they utilized it earlier in the first sentence in the same paragraph. The authors should fix this issue.
Response 6: We appreciate the reviewer’s comment. We added definitions of GlcNAc and other monosaccharides at the beginning of this paragraph (line 118) as follows.
“N-glycans which mainly composed of mannose (Man), N-acetylglucosamine (GlcNAc), N-acetylgalactosamine (GalNAc), galactose (Gal), fucose (Fuc), and sialic acid (N-acetylneumraminic (NeuAc) acid and N-glycolylneuraminic acid (NeuGc) are classified into four groups: high (oligo) mannose type (HM, (Man)5-9(GlcNAc)2); pauci mannose type (PM, (Man)1-4(GlcNAc)2(Fuc)0-1); complex type containing various numbers of Gal, GlcNAc, sialic acid (NeuAc or NeuGc), Fuc and other residues linked to a (Man)3(GlcNAc)2 N-glycan core structure, and hybrid type, which is a hybrid of high mannose and complex oligosaccharide.”
Point 7: In the second sentence (Line 261) in the third paragraph in 3.1. Glycan expression analysis of Npc1 KO CHO cells, “the of unique glycans annotated.” does not make sense. The authors should fix this issue.
Response 7: We appreciate the reviewer’s comment. We corrected the sentence in the second line in the third paragraph in subsection 3.1 (line 339) of revised manuscript as follows.
“Since TAG annotates all glycans with the same m/z values but different structures, we collected only m/z values to represent the unique glycans annotated.”
Point 8: In the seventh and eighth sentences (Lines 267 and 268) in the third paragraph in 3.1. Glycan expression analysis of Npc1 KO CHO cells, some letters were italicized oddly. The authors should fix this issue.
Response 8: We appreciate the reviewer’s comment. We corrected the sentences in line 8 of third paragraph of 3.1 (line 345) of revised manuscript as follows.
“Absolute quantities of glycans measured in both the previous and current study are in good agreement, with correlation coefficients (r2) >0.999. In addition, we generated glycans containing 0-2 acetyl groups for N-glycans and FNGs using TAG List, resulting in two acetylated N-glycans and an acetylated FNG that were not detected in our previous study.”
Point 9: In the eleventh sentence (Lines 272) in the third paragraph in 3.1. Glycan expression analysis of Npc1 KO CHO cells, “provide” should be “provides”.
Response 9: We appreciate the reviewer’s comment. We have corrected the word “provide” in the third lien from the end of last paragraph of 3.1 (line 355) to “provides”.
Point 10: In Figure 8, there is a reference No. 24. That should be No. 26 instead.
Response 10: We appreciate the reviewer’s comment. However, Figure 8 has been removed in accordance with another reviewer’s comment. Therefore, this fix could not be applied.
Point 11: In the figure legend to Figure 10, there is the word “lycomic”. Did the authors mean “glycomic”?
Response 11: We appreciate the reviewer’s comment. Figure 10 has been significantly modified in accordance with the reviewer’s comment in Point 2 (could you see the “Response 2-2” and “Response 2-3”). In this revision, the caption of Figure 10 also changed significantly as follows.
“Figure 10. Representative alterations observed among wt(-), NPC(-), and NPC(+) series using bar charts (upper in each panel) and with mean relative abundance and p-value between wt(-) and NPC(-) (lower) for N-glycans and FNGs; (a) expression levels of HM and PM N-glycans. (b) complex biantennary N-glycans (c) complex triantennary N-glycans (d) complex tetraantennary N-glycans. (e) FNGs possibly derived through the action of pyrophosphatase. (f) FNGs possibly derived through the actions of OST and/or PNGase. (g) complex FNGs.”
Round 2
Reviewer 1 Report
Thank you for addressing the previous concerns. I feel that the responses have improved this manuscript which I now recommend be accepted for publication.